# Long-Term Adherence in Overweight Patients with Obstructive Sleep Apnea and Hypertension—A Pilot Prospective Cohort Study

**DOI:** 10.3390/diagnostics13081447

**Published:** 2023-04-17

**Authors:** Ioana Madalina Zota, Mihai Roca, Maria Magdalena Leon, Corina Dima Cozma, Larisa Anghel, Cristian Statescu, Radu Sascau, Monica Hancianu, Cornelia Mircea, Manuela Ciocoiu, Carmen Marinela Cumpat, Florin Mitu

**Affiliations:** 1Department of Medical Specialties (I), Faculty of Medicine, University of Medicine and Pharmacy “Grigore T. Popa”, 16 University Street, 700115 Iasi, Romania; ioana-madalina.chiorescu@umfiasi.ro (I.M.Z.); maria.leon@umfiasi.ro (M.M.L.); cozma.dima@umfiasi.ro (C.D.C.); larisa.anghel@umfiasi.ro (L.A.); cristian.statescu@umfiasi.ro (C.S.); radu.sascau@umfiasi.ro (R.S.); florin.mitu@umfiasi.ro (F.M.); 2Department of Pharmacognosy, Faculty of Pharmacy, “Grigore T. Popa” University of Medicine and Pharmacy, 16 University Street, 700115 Iasi, Romania; monica.hancianu@umfiasi.ro; 3Department of Pharmaceutical Sciences (II), Faculty of Pharmacy, “Grigore T. Popa” University of Medicine and Pharmacy, 16 University Street, 700115 Iasi, Romania; cornelia.mircea@umfiasi.ro; 4Department of Morpho-Functional Sciences (Pathophysiology), Faculty of Medicine, “Grigore T. Popa” University of Medicine and Pharmacy, 16 Universitatii Street, 700115 Iasi, Romania; manuela.ciocoiu@umfiasi.ro; 5Department of Medical Specialties (III), Faculty of Medicine, University of Medicine and Pharmacy “Grigore T. Popa”, 16 University Street, 700115 Iasi, Romania; 6Academy of Medical Sciences, Ion C. Brătianu Boulevard No 1, 030167 Bucharest, Romania

**Keywords:** obstructive sleep apnea (OSA), positive airway pressure (PAP), adherence, quality of life, anxiety, IPAQ-L, depression

## Abstract

Obstructive sleep apnea (OSA) is associated with increased cardiovascular risk, sedentarism, depression, anxiety and impaired quality of life. The long-term effectiveness of positive airway pressure (PAP) is insufficiently studied and limited by poor patient compliance. The aim of this pilot prospective cohort study was to evaluate long-term adherence in overweight patients with moderate–severe OSA and hypertension and to analyze changes in weight, sleepiness and quality of life. We performed a prospective study that included overweight patients with moderate–severe OSA and hypertension who had not undergone previous PAP therapy. All subjects received a standard physical examination, education regarding lifestyle changes and free PAP therapy for 2 months. After five years, the patients were invited to participate in a telephone-based interview regarding PAP compliance and completed standard questionnaires assessing adherence to medication, physical activity, diet, anxiety and quality of life (QoL). Only 39.58% of the patients were adherent to PAP 5 years (58.42 ± 3.70 months) after being diagnosed with moderate–severe OSA. Long-term PAP use results in sustained weight loss; improved blood pressure control, sleepiness and QOL; and lower anxiety and depression scores. PAP compliance was not associated with a higher level of daily physical activity or a healthier diet.

## 1. Introduction

Affecting 2–4% of the middle-aged population [1], obstructive sleep apnea (OSA) is currently the most prevalent sleep-related disorder, with increasing medical, social and economic implications. OSA causes recurrent episodes of upper airway collapse leading to nocturnal desaturations and sleep fragmentation [2], an unfavorable cardiometabolic profile and neurocognitive impairment [3,4]. Positive airway pressure (PAP) is the gold-standard treatment for moderate and severe OSA, but its long-term effectiveness is severely limited by poor patient compliance and high therapy costs. As part of their cardio-respiratory rehabilitation, patients are advised to lose weight, decrease alcohol intake and avoid dorsal decubitus during sleep. However, data regarding long-term PAP adherence and results in OSA are scarce and divergent, varying largely depending on the population and program characteristics [5,6,7,8,9].

OSA patients frequently exhibit fatigue and muscle weakness during exercise, especially when associated with obesity and excessive daytime sleepiness (EDS) [10,11]. As such, sedentarism is highly prevalent in OSA patients, further increasing their risk of obesity, uncontrolled hypertension (HTN), type 2 diabetes and cardiovascular events [11]. Furthermore, EDS recently emerged as an independent risk factor for nonadherence to blood pressure (BP)-lowering treatment [12]. PAP is generally effective in improving daytime sleepiness [13] and this should improve diet and physical activity (PA), thus promoting weight loss. Sadly, most studies report a lack of significant weight changes after PAP [14] and an inconsistent effect on PA and dietary patterns [15].

Chronic sleep disruption has neuropsychiatric consequences that can affect subjects professionally and personally [16,17]. Impaired health-related quality of life (HR-QoL), depression and anxiety frequently coexist in OSA patients and potentiate each other, forming a vicious circle [18,19,20,21,22]. Subsequently, a comprehensive evaluation of OSA patients must include subjective measures, such as HR-QoL, depression and anxiety scores. The effect of PAP on depression, anxiety [23,24] and QoL [16,25,26] is divergent but more pronounced in women [27].

Data regarding long-term PAP results in OSA are limited and divergent, as only a limited number of studies have addressed long-term results in OSA. As a basis for this study, we presumed that long-term adherence to PAP results in sustained improvement in weight, blood pressure, quality of life, a more active lifestyle and healthier food choices. Our objective was to evaluate long-term adherence to pharmacological and non-pharmacological treatment in overweight patients with moderate–severe OSA and hypertension, and to evaluate changes in weight, sleepiness and quality of life.

## 2. Materials and Methods

### 2.1. Patients

Our study included overweight and hypertensive subjects with newly diagnosed moderate or severe OSA, prior to the initiation of PAP therapy. Weight status was classified by body mass index (BMI), calculated as body weight (kilograms) divided by the height^2^ (meters^2^), as follows: overweight, 25.0–20.9 kg/m^2^; class 1 obesity, 30.0–34.9 kg/m^2^; class 2 obesity, 35.0–39.9 kg/m^2^; class 3 obesity, equal or greater 40 kg/m^2^ [28]. HTN was defined as current antihypertensive treatment, resting systolic or diastolic blood pressure values ≥140 mmHg and ≥90 mmHg, respectively [29]. HTN was classified into grade 1 (systolic blood pressure 140–159 mmHg and/or diastolic blood pressure 90–99 mmHg), grade 2 (systolic blood pressure 160–179 mmHg and/or diastolic blood pressure 100–109 mmHg) and grade 3 (systolic blood pressure ≥180 mmHg and/or diastolic blood pressure ≥110 mmHg) [29]. OSA diagnosis was established in the IIIrd Pneumology Clinic in Iași, by six-channel cardiorespiratory polygraphy (Philips Respironics Alice Night One or DeVilbiss Porti 7), manually scored according to the 3rd International Classification of Sleep Disorders criteria [30]. An Apnea–Hypopnea Index (AHI) of 15–30 events/h and >30 events/h was used to define moderate and severe OSA, respectively. PAP effective pressure autotitration was determined with Philips Respironics DreamStation Auto CPAP or Resmed Airsense 10 Autoset. Daytime sleepiness was evaluated using the standard Epworth scale (ESS) [31]. Quality of life was assessed using the standard European Quality of Life 5 Domain (EQ-5D-5L) questionnaire [32].

Exclusion criteria were central sleep apnea, non-OSA primary sleep disorder, any acute medical conditions in the prior 30 days or any other severe chronic diseases except obesity and hypertension.

### 2.2. Study Design

After OSA diagnosis, all subjects were evaluated in the Cardiovascular Rehabilitation Clinic (Rehabilitation Hospital, Iași, Romania) from January to December 2018. The patients underwent standard physical examination and completed the Epworth and EQ-5D-5L questionnaires. All patients were educated regarding the importance of a healthy lifestyle (nutrition and physical activity) and the need of daily PAP use and received free PAP therapy for 2 months (DreamStation Auto CPAP–Philips Respironics, REMstar Auto C-Flex CPAP–Philips Respironics or AirSense 10 AutoSet CPAP–ResMed). Data regarding our cohort, study protocol and preliminary (2 month) results have been previously published [33,34,35,36,37].

Five years after their initial study, enrollment patients were invited to participate in a telephone-based interview. We collected self-reported information regarding PAP adherence, current weight, most recent blood pressure and pulse values recorded by their primary health care provider, smoking status (never, past or current smokers), alcohol use (daily, weekly, rarely or never), history of diabetes (yes, no, impaired glucose tolerance) and a set of questionnaires: EQ-5D-5L, Epworth, Hill-Bone HBP Compliance to High Blood Pressure Therapy Scale (HB-HBP) [38], International Physical Activity Questionnaire–Long Form (IPAQ-L) [39,40], Rapid Eating Assessment for Participants—Shortened Version (REAP-S) [41], General Anxiety Disorder Assessment (GAD-7) [42] and Patient Health Questionnaire-9 (PHQ-9) [43].

According to self-reported PAP adherence, we divided our study population into two subgroups: adherent and nonadherent patients. Adherence was defined as self-reported daily device usage ≥4 h/night, ≥5 days/week, while nonadherence was defined as self-reported PAP usage <4 h/night or <5 days/week [44]. Our protocol is compatible with the Strengthening the Reporting of Observational Studies in Epidemiology (STROBE) checklist [45] and the EQUATOR guidelines for observational studies [46].

### 2.3. Measurements

#### 2.3.1. Body Measurements and Physical Examination

Height and weight were initially assessed without shoes and with light clothing in the morning. For the follow-up interview, patients were asked to weigh themselves in the same conditions.

Baseline blood pressure (BP) was measured in the morning upon hospital admission with a clinically validated automatic upper arm blood pressure monitor (Omron M2), using the Omron Intelliwrap cuff, adequate for arm circumferences between 22 and 42 cm. For the follow-up interview, patients were asked to provide the most recent blood pressure values recorded by their primary health care provider.

Baseline heart rate (HR) was obtained from the electrocardiogram, routinely recorded the day of hospital admission. For the follow-up interview, patients were asked to provide the most recent pulse value recorded by their primary health care provider.

#### 2.3.2. EQ-5D-5L, Epworth, HB-HBP, IPAQ-L, REAP-S, GAD-7 and PHQ-9 Questionnaires

The Epworth questionnaire [31] was completed before PAP initiation and repeated during the follow-up interview, with the help of the same certified health care provider who could offer supplementary information, when necessary. The global ESS score was recorded both times. Somnolence was classified as normal (ESS score 0–10), mild (ESS score 11–12), moderate (ESS score 13–15) and severe (ESS score 16–24).

The EQ-5D-5L questionnaire was completed before PAP initiation and repeated during the follow-up interview, with the help of the same certified health care provider who could offer supplementary information, when necessary. The first part of the questionnaire rates 5 domains (mobility, self-care, usual activities, pain/discomfort and symptoms of anxiety/depression) on a scale of 1 to 5 (no problem, slight, moderate, severe and extreme problems, respectively). For the second part of the questionnaire the patient is asked to provide a self-assessed level of current health, using a visual analogue scale labeled from 0 (the worst imaginable health status) to 100 (the best possible health status). Utility index values can be calculated in order to ease the interpretation of the first part of the HR-QoL. We calculated the EQ-5D-5L Index Value using the recently published value sets for Romania [47].

The IPAQ-L was completed during the follow-up interview. The total time spent performing various types of PA was converted into minutes. Physical activity levels are expressed by MET-min/week, using the following formula: MET level × minutes of activity/day × days per week. In line with IPAQ-L scoring guidelines, MET level is a predefined coefficient attributed to each type of PA. We calculated the total MET-min/week and the corresponding MET value per activity (walking, moderate and vigorous effort) and per domain (work, transportation, leisure and domestic and garden). IPAQ-L results differentiate three levels of physical activity: high (at least 3 days of vigorous PA accumulating ≥1500 MET-min/week OR daily PA accumulating at least 3000 MET-minutes/week), moderate (at least 3 days of vigorous-intensity activity of at least 20 min/day OR at least 5 days of moderate PA/walking of at least 30 min/day OR at least 5 days of any combination of PA achieving ≥600 MET-min/week) and low (does not meet the criteria for moderate/high PA) [48].

Adherence to pharmacological and non-pharmacological treatment for HBP was estimated using Hill-Bone HBP Compliance to High Blood Pressure Therapy Scale (HB-HBP). The questionnaire has 9 items assessing medication adherence, 3 items assessing sodium intake and 2 items assessing appointment keeping subscale. Each item is scored on a 4-point Likert scale, a score of 4 corresponding to the highest level of adherence. 

The REAP-S questionnaire was completed according to the previous week’s intake. Response options for each question range from 1 to 3 on a 3-point Likert-scale: 1 = usually/often, 2 = sometimes, 3 = rarely/never or does not apply to me. The total score was calculated by summing the individual responses for the first 13 questions, a higher score indicating a healthier diet.

We evaluated the presence of anxiety symptoms with the 7-item GAD-7 scale. Response options for each question range from 0 to 3 on a 4-point Likert-scale: 0 = not at all, 1 = several days, 2 = more than half of the days, 3 = nearly every day. The total GAD-7 score was calculated by summing the individual responses for all 7 items. Due to low GAD-7 scores in our study, we considered a score ≥5 “with anxiety”.

We evaluated the presence of depression symptoms with the PHQ-9 questionnaire. The scale consists of 9 items exploring symptoms experienced in the 2 preceding weeks. Response options for each question range from 0 to 3 on a 4-point Likert-scale: 0 = not at all, 1 = several days, 2 = more than half of the days, 3 = nearly every day. The total PHQ-9 score was calculated by summing the individual responses for all 9 items. Due to low PHQ-9 scores in our study, we considered a score ≥5 “with depression”.

#### 2.3.3. Telephone-Based Interview

PAP usage was dichotomized as “yes” or “no”. Patients who had stopped PAP were asked to provide the reason for PAP withdrawal. Patients who continued PAP therapy were asked to provide an estimate for device usage—average hours/night and average days/week.

### 2.4. Statistical Analysis

Data analysis was conducted using SPSS 20.0 (Statistical Package for the Social Sciences, Chicago, Illinois). Data were presented as mean ± standard deviation (SD), as median with interquartile range and as number of cases with percent frequency, for continuous normally distributed, non-normally distributed and categorical variables, respectively. Categorical comparisons were performed by Chi-square test or by Fisher’s exact test. Continuous normally distributed variables were compared by Independent Samples *t*-test. The nonparametric Mann–Whitney U test was applied when the distribution of continuous variables did not satisfy the assumption of normality. 

The dependent *T*-test (paired-samples *T*-test) was used to compare the means between two related groups on the same continuous, normally distributed variable. Paired-samples comparisons of non-normally distributed variables were performed applying Wilcoxon nonparametric test. 

Multivariate logistic regression analysis was performed to assess the independent predictors of PAP compliance. A two-sided *p* value < 0.05 was considered significant for all analyses.

The study protocol received ethical approval by the Ethics Committee of the Grigore T. Popa University of Medicine and Pharmacy in Iași (ethical approval codes 1183/17.01.2018 for the baseline evaluation and 273/20.02.2023 for the follow-up) and complied with the Declaration of Helsinki [49]. All patients signed a written informed consent for the initial evaluation and gave verbal consent for the telephone-based interview, 5 years after the initial study enrollment. 

## 3. Results

Of the 154 patients referred to our sleep clinic between January and December 2018, 72 met the inclusion criteria. A total of 24 patients were lost during the follow-up (Figure 1). Our final study group included 48 patients aged 61.66 ± 9.03 years who participated in the telephonic follow-up 58.42 ± 3.70 months after their initial study enrollment. The majority of our patients were male (64.58%), non-smokers (50%) and suffered from severe OSA (62.50%). The demographic characteristics of the study group are illustrated in Figure 2 and Table 1.

Our two subgroups (adherent and nonadherent to PAP) were homogenous regarding the baseline clinical parameters, quality of life and daytime sleepiness, *p* > 0.05 (Table 1).

The PAP adherence in our study group after 58.42 ± 3.70 months was 39.58% (Figure 2). The main reason for PAP discontinuation was economical (44.83%), followed by device intolerance due to side effects (37.94%) (Figure 3). A multivariate logistic regression analysis did not find AHI, age, BMI and quality of life to be significant predictors for long-term PAP adherence in our study group (*p* > 0.05).

While the PAP-adherent patients presented a significant decrease in weight and BMI (Figure 4 and Figure 5), the nonadherent patients exhibited a tendency toward weight gain that did not reach statistical significance (*p* > 0.05) (Table 2). The HR increased in PAP-nonadherent patients (Table 2). While the DBP decreased in patients who continued to use PAP (Figure 6, Table 2), both the SBP and DBP increased in those who discontinued the positive airway treatment (Table 2).

Although the median baseline ESS score was 8, corresponding to “normal” daytime somnolence, we observed a statistically significant improvement in daytime sleepiness both in the adherent and nonadherent patients (Table 2, Figure 7).

According to the EQ-5D-5L questionnaire, most of the subjects reported no difficulties with movement, self-care, daily activities and anxiety or depression. The EQ-5D-5L index significantly improved only in PAP-adherent patients (Table 2). The median VAS scale reflecting subjective quality of life also improved, with borderline statistical significance, only in the PAP-adherent patients (Table 2).

PAP discontinuation was associated with higher GAD-7 scores (*p* = 0.003). The association remained significant for items 1, 3, 4 and 6 (feeling nervous/anxious/on edge, worrying too much about different things, trouble relaxing and becoming easily annoyed/irritable, respectively) (Table 3). Anxiety was more prevalent in PAP-nonadherent (31.03%) versus -adherent (10.52%) patients (*p* < 0.001).

PAP nonadherence was significantly associated with overall higher PHQ9 scores (*p* = 0.007). The association remained significant for the self-reported sleep difficulties (*p* = 0.003), lack of energy/tiredness (*p* = 0.002) and appetite disruptions (*p* = 0.018) items (Table 3). Depression was more prevalent in PAP-nonadherent (37.93%) versus -adherent (5.26%) patients (*p* < 0.001).

The total PA, PA per each analyzed domain (work, leisure, domestic and transport) and REAP total score and per each item were similar in the 2 subgroups (*p* > 0.05) (Table 3). PAP nonadherence was significantly associated with lower HB-HBP scores (*p* = 0.008), that remained statistically significant only in the “reduced sodium” subset (*p* = 0.003) (Table 3).

## 4. Discussion

This five-year prospective follow-up study showed that long-term PAP in OSA patients results in sustained weight loss, better BP control, improved sleepiness and QOL as well as lower anxiety and depression scores. On the other hand, PAP compliance was not associated with a more active lifestyle or a healthier diet. 

PAP remains the gold-standard treatment for moderate–severe OSA, but its effectiveness largely depends on regular device use. Despite technical progress in PAP pressure delivery modes and mask materials and design, PAP compliance remains poor and has not improved in the past 2 decades [50], as 8–46% of subjects discontinue using the device within the first 5 years [51]. Elevated nasal resistance increases expiratory discomfort and lowers PAP adherence but is usually overlooked in clinical practice [52]. Compared to other demographic and clinical variables, apnea severity (AHI) seems to be the most robust predictor of long-term PAP adherence [50]. However, older patients [53], with higher ESS scores [54], tend to be more compliant to PAP [53].

The 5-year PAP compliance rate in our study group was 39.58%. The primary reason for PAP discontinuation was financial difficulty (44.83%), followed by device-related side effects (mask leak, mucosal irritation or difficulty while exhaling) (37.94%). PAP cost is not covered by the Romanian public health care system and is high when compared to the country’s average income. However, other factors, especially inadequate device pressure, mask-related discomfort, stigma and shame, significantly diminish PAP adherence [33]. The economical factor could explain why AHI, age, BMI and quality of life were not significant predictors for long-term PAP adherence in our study. Consequently, we underline the need for OSA support groups and dedicated medical education programs as well as health care economic support measures to improve PAP adherence [55].

The presence of daytime sleepiness is a predictor of severe apnea and of a higher clinical benefit with PAP use [56]. The baseline median ESS scores in our OSA patients were within the “normal” range which could also explain the poor long-term PAP adherence and the lack of association between apnea severity, age and BMI with long-term PAP use. However, in line with previous studies [10,27,53,57], the ESS scores significantly decreased after PAP therapy, especially in the adherent patients. As Epworth scores are correlated with BMI [58], this effect can be partly attributed to the weight changes observed in the PAP-compliant subgroup.

PAP regulation of the neurovegetative and hormonal determinants of HTN explains its beneficial effect on BP levels and the circadian pattern, especially in patients with resistant hypertension [59,60]. However, a recent report also noted improvement in BP values in normotensive patients, suggesting that PAP adherence could modulate the baroreflex threshold [61]. Furthermore, PAP therapy seems to have a more robust effect on the diastolic BP value [61], which is consistent with our findings.

The long-term PAP results in OSA have been addressed by a limited number of studies. A surgical approach documented significant long-term (5 years) improvements in sleep parameters and EDS [62]. A 1-year retrospective follow-up of 121 patients showed that improvement in sleep quality, ESS and BP occurs only in PAP-adherent patients [7]. After 2 years, PAP use is associated with a significant decrease in DBP but not in weight [61], and a recent 5-year follow-up study found that PAP significantly reduces cerebrovascular and hypertensive event incidence but not coronary event risk [8]. Although medium-term (6 months) PAP use can prevent further arterial stiffening in patients with OSA and resistant hypertension [63], a 7.5-year follow-up observational study showed that PAP compliance does not prevent arterial stiffness progression in obese OSA patients [64]. Interestingly, two previous reports analyzed long-term PAP use in Romanian patients. A 30-month follow-up of 7 patients with OSA and non-resistant HTN documented significant SBP and DBP changes but no change in BMI [65]. On the other hand, and consistent with our results, a 4-year follow-up of Romanian patients with OSA and resistant hypertension associated PAP adherence with sustained weight loss [9]. Obesity and OSA are in an interdependence mechanism, forming a vicious circle—while excess weight is a leading cause of OSA, the latter promotes supplementary weight gain by decreased energy expenditure and hormonal dysregulation of the hunger–satiety mechanisms [66]. Although PAP adherence partially reverses these hormonal changes [67,68], it also reduces energy expenditure [69], which can promote weight gain in the absence of a compensatory increase in PA [15]. Weight gain seems to be directly associated with depressive symptoms and baseline BMI [57]. On the other hand, decreased rapid eye movement sleep is associated with reduced resting metabolic rate and unhealthy food choices [70] and PAP adherence improves leptin resistance and ghrelin plasma levels, favoring appetite control [67,68]. In the European Sleep Apnea Database Cohort, PAP was associated with a reduction in BMI only in obese patients with high baseline ESS scores [71]. We noted a statistically significant change in BMI only among the PAP-adherent patients, more substantial than that reported by Pleava et al. [9]. This could be explained by the longer follow-up duration of our patients and by the fact that our initial evaluation took place in a cardiovascular rehabilitation clinic where the patients were educated regarding the importance of nutrition and exercise. In order to achieve maximal risk factor modification in OSA patients, PAP therapy should be combined with active weight reduction programs that encompass both exercise training and nutritional counseling.

Unfortunately, PA levels in OSA patients are negatively associated with OSA severity [72] and are lower than in the general population [3,73]. The primary limitation of self-reporting PA questionnaires is the subject’s ability to recollect recent activities and classify them as light, moderate and/or vigorous [74]. Like most self-reporting PA questionnaires, the IPAQ-L tends to overestimate moderate–vigorous PA and underestimate sitting time [75]. However, IPAQ-L results are closer to the real PA level when completed in the presence of a trained medical professional [75], and a recent study reported a significant association between the IPAQ results and actual PA (pedometer steps) [53]. Impaired exercise capacity in OSA is caused by physical limitations (dyspnea and muscular fatigue due to neurohormonal, mitochondrial and metabolic disturbances) [76,77] but also by a lack of psychological motivation, secondary to daytime sleepiness and fatigue [72]. Although three previous studies associated short- and medium-term PAP therapy with a progressive increase in individual PA [53,78,79], in two other reports, three months of PAP was not associated with increased PA [80,81]. Our analysis showed that patients with moderate–severe OSA have low PA levels, irrespective of PAP adherence. It was postulated that improved daytime sleepiness is what favors a more active lifestyle [53]. Our patients’ low baseline ESS scores could explain the similar self-reported PA between our adherent and nonadherent subgroups.

Decreased PA can also be attributed to anxiety and depression [73]. Repetitive hypoxia causes neural injury, fatigue, impaired concentration and loss of libido [81]. This explains behavioral changes in pediatric OSA patients [82] and that impaired QoL, depression and anxiety, which frequently coexist and potentiate each other in adult OSA patients [57,73,81], are partially reversed by PAP [81]. PAP improves sleep quality [27,57], which partially explains our lower PHQ-9 score results (self-reported sleep difficulties (*p* = 0.003) and lack of energy/tiredness (*p* = 0.002)) in adherent patients. In line with previous reports [83], anxiety and depression were significantly more prevalent in patients who had discontinued PAP therapy (*p* < 0.001). HR-QoL illustrates the influence of disease on every-day life from the patient’s perspective. Although a subjective concept *per se*, standardized questionnaires such as EQ-5D-5L transform QoL into a research appropriate parameter. While some studies suggested that PAP “pseudo-normalizes” QoL up to the level of healthy controls, in other reports QoL did not significantly improve after positive airway therapy [16,25,26]. Our results show that QoL is impaired in Romanian OSA patients and is partially corrected after long-term PAP.

Although decreased REM sleep is associated with unhealthy food choices [70], long-term PAP use does not seem to alter dietary patterns [15]. This is in line with our results, as the REAP-S results did not significantly vary according to PAP use. However, the “reduced sodium” subset of the HB-HBP questionnaire showed that PAP-compliant patients were more adherent to the low-sodium diet.

The association between OSA and uncontrolled BP values could partly be explained by nonadherence to medication and lifestyle changes [12]. Impaired concentration, forgetfulness and anxiety, frequently reported by OSA patients, are considered major triggers for nonadherence in hypertensive patients [84]. Sleepiness contributes to nonadherence and EDS recently emerged as an independent risk factor for nonadherence to BP-lowering therapy [12]. However, our moderate–severe OSA patients reported very good HB-HBP adherence scores, irrespective of long-term PAP compliance. These findings could be explained by the low baseline and follow-up ESS scores recorded in our OSA patients.

Our results have clinical and practical significance. A major finding of our analysis is similar PA levels and nutritional choices in adherent and nonadherent patients, despite differences regarding QoL and psychological status. Further studies are needed to understand the reasons for limited PA in adherent OSA patients. These patients should not be viewed as “compliant”—sustained efforts should be made in order to promote adherence to lifestyle changes in all OSA patients. We emphasize the need for the development of comprehensive cardio-pulmonary rehabilitation programs to address both somatic and psychological OSA consequences and correct modifiable cardiovascular risk factors [85]. Another key result is the importance of the financial factor in the decision to discontinue PAP. Governmental financial policies regarding PAP use (device funding and health care coverage) could significantly increase treatment compliance, especially in low- and middle-income countries.

The strength of the study is the long-term follow-up of OSA patients, allowing a dynamic evaluation of weight status, comorbidities, ESS score and quality of life. We applied a comprehensive number of validated questionnaires to assess adherence to medication, lifestyle and psychological status. The study is limited by its few participants and low statistical power (69%), which limits the generalizability of our results. Another important limitation is the collection of follow-up data via telephone, which raises the concern regarding the validity of patient-reported clinical data (weight status, blood pressure and heart rate) and questions the correct assessment of PAP adherence (hours/night), which was essential for the study group categorization (adherent versus nonadherent).

## 5. Conclusions

Less than 40% of patients are adherent to PAP 5 years after being diagnosed with moderate–severe OSA. Long-term PAP use could promote sustained weight loss, improved DBP, sleepiness and QOL as well as lower anxiety and depression scores, but PAP compliance does not seem to impact daily PA or diet. The implementation of comprehensive cardio-pulmonary rehabilitation programs and governmental financial policies could increase adherence to PAP and lifestyle changes in OSA.

## Figures and Tables

**Figure 1 diagnostics-13-01447-f001:**
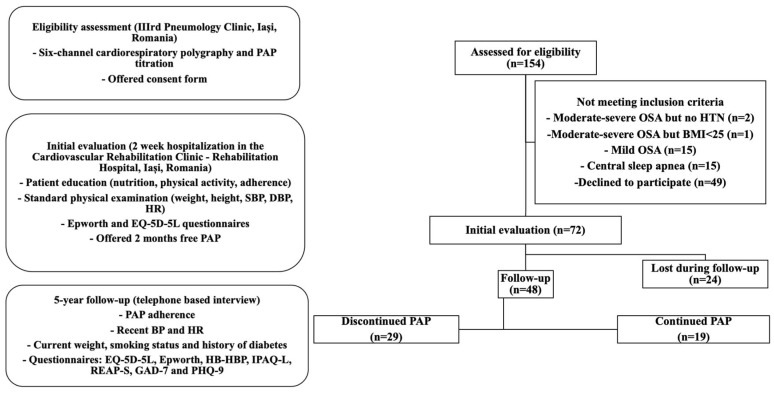
Flowchart diagram of patients referred to our sleep clinic January to December 2018. OSA: obstructive sleep apnea; HTN: hypertension; BMI: body mass index; PAP: positive airway pressure; SBP: systolic blood pressure; DBP: diastolic blood pressure; HR: heart rate; EQ-5D-5L: European Quality of Life 5 Domain questionnaire; HB-HBP: Hill-Bone HBP Compliance to High Blood Pressure Therapy Scale; IPAQ-L: International Physical Activity Questionnaire—Long Form; REAP-S: Rapid Eating Assessment for Participants—Shortened Version; GAD-7: General Anxiety Disorder Assessment; PHQ-9: Patient Health Questionnaire-9.

**Figure 2 diagnostics-13-01447-f002:**
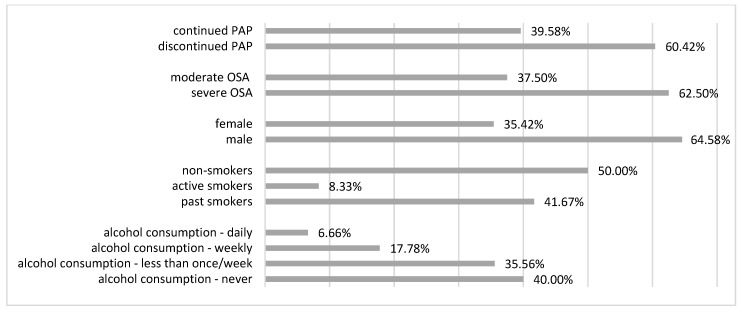
Follow-up demographic and clinical characteristics of our study group. PAP: positive airway pressure; OSA: obstructive sleep apnea.

**Figure 3 diagnostics-13-01447-f003:**
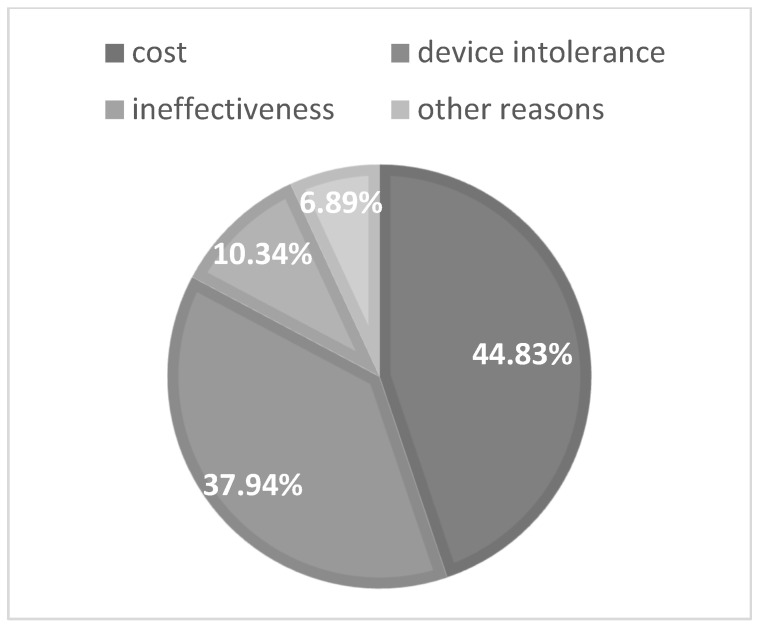
Reasons for PAP withdrawal. PAP: positive airway pressure.

**Figure 4 diagnostics-13-01447-f004:**
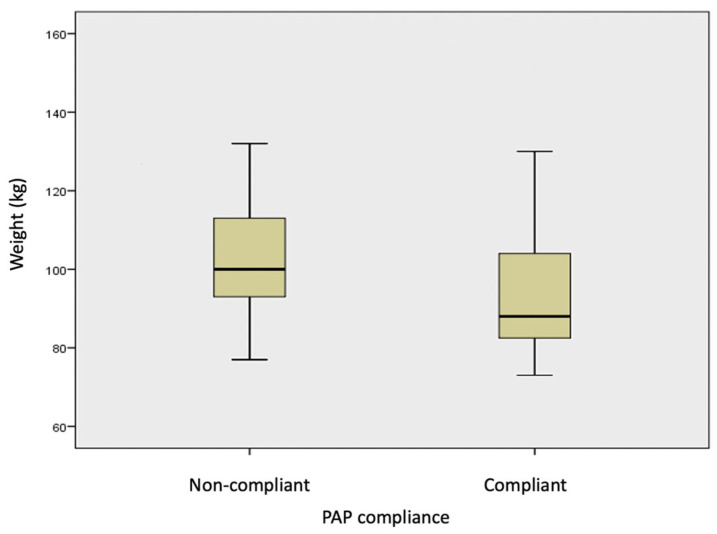
Changes in weight—baseline versus follow-up. PAP: positive airway pressure.

**Figure 5 diagnostics-13-01447-f005:**
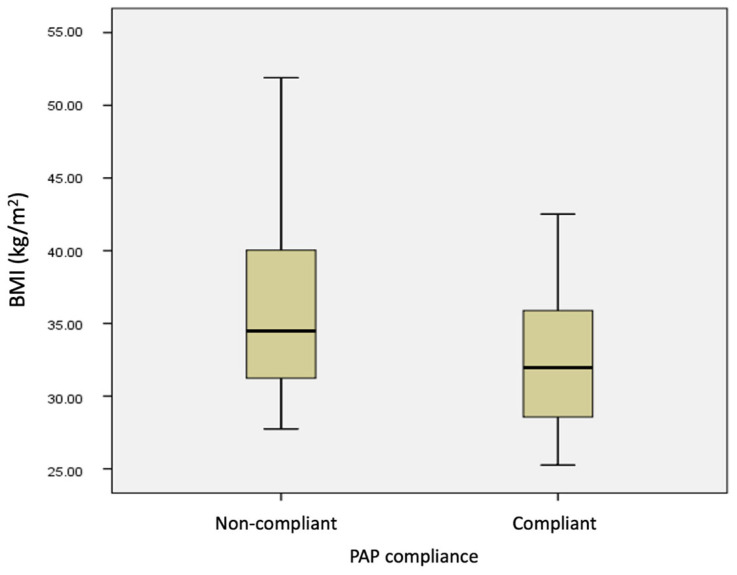
Changes in BMI—baseline versus follow-up. BMI: body mass index; PAP: positive airway pressure.

**Figure 6 diagnostics-13-01447-f006:**
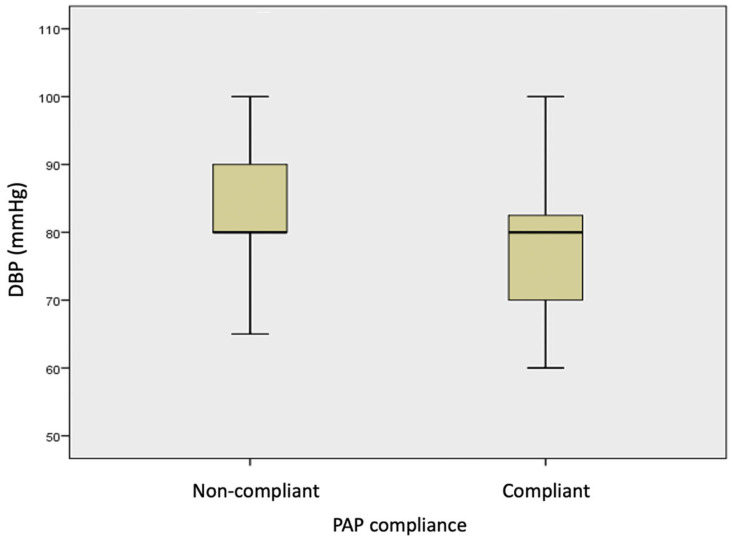
Changes in DBP—baseline versus follow-up. DBP: diastolic blood pressure; PAP: positive airway pressure.

**Figure 7 diagnostics-13-01447-f007:**
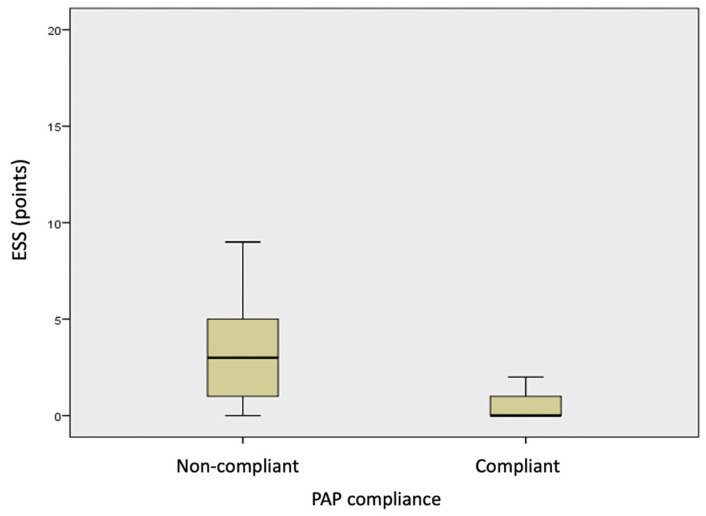
Changes in ESS—baseline versus follow-up. ESS: Epworth sleep score results; PAP: positive airway pressure.

**Table 1 diagnostics-13-01447-t001:** Baseline clinical, anthropometric, quality of life and daytime sleepiness parameters.

	PAP Nonadherent (*n* = 29)	PAP Adherent (*n* = 19)	*p* Value
Age (years)	61.14 ± 8.95	62.47 ± 9.34	0.776
AHI (events/hour)	39.47 ± 19.71	42.69 ± 19.34	0.562
Weight (kg)	103.14 ± 17.29	99.79 ± 19.73	0.411
BMI (kg/m^2^)	35.47 ± 5.78	34.20 ± 4.81	0.417
HR (bpm)	72.48 ± 10.21	75 ± 10.55	0.41
SBP (mmHg)	133.93 ± 18.96	142.05 ± 19.95	0.16
DBP (mmHg)	84.52 ± 13.79	91.68 ± 9.57	0.05
EQ-5D-5L index	0.86 (0.77–0.95)	0.94 (0.82–1)	0.362
EQ-5D-5L-VAS	60 (50–77.5)	75 (55–87)	0.115
ESS (points)	8 (4–13)	8 (4.5–10)	0.546

OSA: obstructive sleep apnea; BMI: body mass index; AHI: apnea–hypopnea index; HR: heart rate; SBP: systolic blood pressure; DBP: diastolic blood pressure; EQ-5D-5L: European Quality of Life 5 Domain questionnaire; VAS: visual analog scale; ESS: Epworth sleepiness scale.

**Table 2 diagnostics-13-01447-t002:** Comparative analysis of clinical parameters, quality of life and daytime sleepiness, in PAP-adherent and -nonadherent groups.

	PAP-Nonadherent Baseline	PAP-Nonadherent Follow-Up	*p* %	PAP-Adherent Baseline	PAP-Adherent Follow-Up	*p* #
Weight (kg)	103.14 ± 17.29	104.14 ± 19.08	0.38	99.79 ± 19.73	94.79 ± 17.84	0.01
BMI (kg/m^2^)	35.41 ± 5.78	35.88 ± 5.93	0.29	34.20 ± 4.81	32.43 ± 4.56	0.005
HR (bpm)	72.48 ± 10.21	76.83 ± 11.61	0.04	75 ± 10.55	70.05 ± 12.06	0.20
SBP (mmHg)	133.93 ± 18.96	136.79 ± 9.92	0.49	142.05 ± 19.95	135.37 ± 13.19	0.18
DBP (mmHg)	84.52 ± 13.79	81.86 ± 11.51	0.37	91.68 ± 9.57	78.95 ± 9.41	<0.001
EQ-5D-5L item 1 (points)	1 (1–2)	2 (1–3)	0.17	1 (1–1)	1 (1–1)	0.33
EQ-5D-5L item 2 (points)	1 (1–2)	1 (1–1)	0.14	1 (1–1)	1 (1–1)	0.18
EQ-5D-5L item 3 (points)	1 (1–2)	1 (1–2)	0.49	1 (1–1)	1 (1–1)	0.10
EQ-5D-5L item 4 (points)	3 (1–3)	2 (1–3)	0.36	2 (1–3)	1 (1–2)	0.13
EQ-5D-5L item 5 (points)	2 (1–3)	1 (1–2.5)	0.54	1 (1–2)	1 (1–1)	0.006
EQ-5D-5L index	0.86 (0.77–0.95)	0.90 (0.79–0.96)	0.98	0.94 (0.82–1)	0.96 (0.92–1)	0.03
EQ-5D-5L VAS	60 (50–77.5)	60 (50–75)	0.39	75 (55–87)	80 (75–90)	0.06
ESS (points)	8 (4–13)	3 (0.5–5)	<0.001	8 (4.5–10)	0 (0–1)	<0.001

PAP: positive airway therapy; BMI: body mass index; HR: heart rate; SBP: systolic blood pressure; DBP: diastolic blood pressure; EQ-5D-5L: European Quality of Life 5 Domain questionnaire; VAS: visual analog scale; ESS: Epworth sleepiness scale; %: baseline-follow-up comparison in nonadherent patients; #: baseline-follow-up comparison in adherent patients.

**Table 3 diagnostics-13-01447-t003:** Comparative analysis of anxiety, depression, diet, adherence and physical activity in PAP-adherent and -nonadherent patients.

	All Patients (*n* = 48)	PAP Nonadherent (*n* = 29)	PAP Adherent (*n* = 19)	*p* Value (Adherent vs. Nonadherent)
GAD-7 total (points)	1 (0–4.5)	3 (1–5.25)	0 (0–1)	0.003
Item 1 (points)	0 (0–0.1)	1 (0–1)	0 (0–0)	0.004
Item 2 (points)	0 (0–0.5)	1 (0–1)	0 (0–0)	0.22
Item 3 (points)	0 (0–1)	1 (0–1)	0 (0–0)	0.004
Item 4 (points)	0 (0–1)	1 (0–1)	0 (0–0)	0.01
Item 5 (points)	0 (0–0)	1 (0–1)	0 (0–0)	0.06
Item 6 (points)	0 (0–1)	0.5 (0–1)	0 (0–0)	0.005
Item 7 (points)	0 (0–1)	0 (0–1.25)	0 (0–0)	0.08
PHQ-9 total (points)	3 (0.5–5)	4 (2–6.25)	1 (0–3)	0.005
Item 1 (points)	0 (0–0)	0 (0–0)	0 (0–0)	0.16
Item 2 (points)	0 (0–1)	0.5 (0–1)	0 (0–0)	0.05
Item 3 (points)	1 (0–2)	1 (0.75–2.25)	0 (0–1)	0.001
Item 4 (points)	1 (0–1)	1 (0–2)	0 (0–1)	0.001
Item 5 (points)	0 (0–0)	0 (0–1)	0 (0–0)	0.009
Item 6 (points)	0 (0–0)	0 (0–0)	0 (0–0)	0.16
Item 7 (points)	0 (0–0.5)	0 (0–0.25)	0 (0–0)	0.87
Item 8 (points)	0 (0–0)	0 (0–0.25)	0 (0–1)	0.39
Item 9 (points)	0 (0–0)	0 (0–0)	0 (0–0)	1
REAP-S (points)	30 (28–32)	30 (29–32.75)	29 (27–32)	0.84
HB-HBP total (points)	52 (50–53)	51 (49–53)	53 (51.75–54)	0.004
HB-HBP reduced sodium (points)	11 (10–11.5)	10 (9–11)	11 (11–12)	0.003
HB-HBP appointment (points)	6 (6–6.5)	6 (5–7)	6 (6–6.25)	0.28
HB-HBP medication (points)	36 (34.5–36)	35 (34–36)	36 (35–36)	0.11
IPAQ-L–Total MET	2542.5 (816.75–4494.13)	3102 (893.5–4729.5)	2002.5 (816–3693)	0.56
IPAQ-L–MET work	0 (0–0)	0 (0–0)	0 (0–0)	0.47
IPAQ-L–MET transport	453.75 (74.25–1188.00)	462 (82.5–1188)	445.5 (0–1155)	0.71
IPAQ-L–MET domestic	720 (180–1957.5)	1140 (195–2475)	720 (180–1170)	0.25
IPAQ-L–MET leisure	239 (0–823.5)	292 (16.5–816)	120 (0–834)	0.38
IPAQ-L–MET walking	841.5 (222.75–1658.25)	792 (247.5–1600.5)	1039.5 (198–1881)	0.68
IPAQ-L–MET moderate	975 (375–2970)	1770 (275–3195)	720 (540–1860)	0.34
IPAQ-L–MET vigorous	0 (0–0)	0 (0–0)	0 (0–0)	0.73

PAP: positive airway therapy; BMI: body mass index; HR: heart rate; SBP: systolic blood pressure; DBP: diastolic blood pressure; EQ-5D-5L: European Quality of Life 5 Domain questionnaire; VAS: visual analog scale; ESS: Epworth sleepiness scale; GAD-7: General Anxiety Disorder Assessment; PHQ-9: Patient Health Questionnaire-9; REAP: Rapid Eating Assessment for Participants—Shortened Version; HB-HBP: Hill-Bone HBP Compliance to High Blood Pressure Therapy Scale; IPAQ-L: International Physical Activity Questionnaire—Long Form; MET: metabolic equivalents.

## Data Availability

The data that support the findings of this study are available upon request from the corresponding authors.

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
