# Peer review of "Long-Term Adherence in Overweight Patients with Obstructive Sleep Apnea and Hypertension—A Pilot Prospective Cohort Study"

_diagnostics, 2023, doi:10.3390/diagnostics13081447_

Round 1
Reviewer 1 Report
I am glad to review this interesting work which addressed several important issues regarding OSA. My comments are shown as below:
Major concerns:
1. The adherence to PAP was determined by telephone interview. This made study group categorization (adherent vs. non-adherent) not so exact, resulting in the findings and conclusions questionable.
2. I suggest to add clinical implications of the study findings in the "Discussion" and " Conclusions". This could make this work more informative to the readers.
Minor comments:
1. Line 74, the full stop following hypertension should be comma.
2. Lines 80-82, One reference should be cited for the obesity classification.
3. Units should be given in the Tables and Figures.
4. Line 293, a hyphen should be given between "follow" and "up".
5. Lines 297-300, male predominance is the inherent finding in all studies on OSA. Thus, the discussion in this paragraph is nonsense. So I suggest to delete this paragraph.
6. Lines 301-309, please explain why these parameters, including AHI, age, BMI, and QOL, could not predict long-term compliance of PAP in the present study. Is it because of fewer participants?
Author Response
We sincerely thank the reviewer for his/her thoughtful comments and constructive suggestions. We have revised our manuscript in light of these recommendations and completed our manuscript with new information, as detailed below.
- The adherence to PAP was determined by telephone interview. This made study group categorization (adherent vs. non-adherent) not so exact, resulting in the findings and conclusions questionable.
We modified our limitations paragraph emphasizing the importance of PAP adherence assessment.
The study is limited by its few participants and potentially low statistical power, which limits generalizability of our results. Another important limitation is the collection of follow-up data via telephone, which raises the concern regarding the validity of patient reported clinical data (weight status, blood pressure, heart rate) and questions the correct assessment of PAP adherence (hours/night), which was essential for the study group categorization (adherent versus non-adherent).
- I suggest to add clinical implications of the study findings in the "Discussion" and " Conclusions". This could make this work more informative to the readers.
We completed our manuscript as follows
Discussion
Our results have clinical and practical significance. A major finding of our analysis is similar PA levels and nutritional choices in adherent and non-adherent patients, despite differences regarding QoL and psychological status. Further studies are needed to understand the reasons for limited PA in adherent OSA patients. These patients should not be viewed as ”compliant” - sustained efforts should be made in order to promote adherence to lifestyle changes in all OSA patients. We emphasize the need for development of comprehensive cardio-pulmonary rehabilitation programs to address both somatic and psychological OSA consequences and correct modifiable cardiovascular risk factors [85]. Another key result is the importance of the financial factor in the decision to discontinue PAP. Governmental financial policies regarding PAP use (device funding and healthcare coverage) could significantly increase treatment compliance, especially in low-and middle-income countries.
Conclusions
Less than 40% of patients are adherent to PAP 5 years after being diagnosed with moderate-severe OSA. Long term PAP use could promote sustained weight loss, improved DBP, sleepiness and QOL as well as lower anxiety and depression scores, but PAP compliance does not seem to impact daily PA or diet. Implementation of comprehensive cardio-pulmonary rehabilitation programs and governmental financial policies could increase adherence to PAP and lifestyle changes in OSA.
Minor comments:
- Line 74, the full stop following hypertension should be comma.
We apologize for the mistake, we performed the minor correction.
- Lines 80-82, One reference should be cited for the obesity classification.
We added an appropriate reference:
Weight status was classified by body mass index (BMI) - calculated as body weight (kilograms) divided by the height2(meters2), as follows: overweight 25.0–20.9 kg/m2, class 1 obesity - 30.0–34.9 kg/m2, class 2 obesity - 35.0–39.9 kg/m2, class 3-obesity - equal or greater 40 kg/m2 [28].
- Units should be given in the Tables and Figures.
We completed the tables and figures with the missing units.
- Line 293, a hyphen should be given between "follow" and "up".
We apologize for the mistake, we performed the minor correction.
- Lines 297-300, male predominance is the inherent finding in all studies on OSA. Thus, the discussion in this paragraph is nonsense. So I suggest to delete this paragraph.
We deleted the unnecessary paragraph.
- Lines 301-309, please explain why these parameters, including AHI, age, BMI, and QOL, could not predict long-term compliance of PAP in the present study. Is it because of fewer participants?
We discussed this matter, as follows:
The economical factor could explain why AHI, age, BMI and quality of life were not significant predictors for long-term PAP adherence in our study.
And
Baseline median ESS scores in our OSA patients were within ”normal” range which could also explain poor long-term PAP adherence and lack of association between apnea severity, age and BMI with long-term PAP use.
We hope that you find our responses satisfactory and that we have managed to improve the quality of our manuscript,
Sincerely,
All authors
Reviewer 2 Report
Dear authors the paper is interesting, well done. However perform the corrections required:
- Obstructive sleep apnea (OSA) is associated with cardiovascular co-morbidities and mortality. Arterial stiffness is an independent predictor of cardiovascular risk and mortality, and is influenced by the presence of OSA and related comorbidities. There is a paucity of data regarding long-term evolution of arterial stiffness in CPAP-treated OSA patients., please discuss and cite doi:10.1371/journal.pone.0236667.
- please better explain study protocol via a flow-chart
- In the adherence to treatment of the patient with obstructive sleep apnea, mood or behavioral disorders of the patient are molot relevant to consider. Indeed, it has been shown how these disorders are already present in the child and therefore should be well investigated. Likewise, the disorders are reduced with surgical or cpap therapy., please discuss and cite doi:10.3390/children8100921
-please adopt the adequate guidelines to improve the research protocol according to the equator database.
- Resistant hypertension (RHT) is associated with obstructive sleep apnea (OSA) and increased aortic stiffness, measured by carotid-femoral pulse wave velocity (cf-PWV). An interesting study evaluated in a randomized controlled trial, the effect of Continuous positive airway pressure (CPAP) treatment on cf-PWV in comparison with a control group in patients with RHT and moderate-severe OSA. One-hundred and sixteen patients were randomized to 6-month CPAP treatment (56 patients) or no therapy (60 patients), while keeping their antihypertensive treatment unchanged. Carotid-femoral pulse wave velocity was performed at the beginning and end of the 6-month period. Intention-to-treat intergroup differences in cf-PWV changes were assessed by a generalized mixed-effects model with the allocation group as a fixed factor and adjusted for age, sex, changes in mean arterial pressure and the baseline cf-PWV values. In conclusion, a 6-month CPAP treatment did not reduce aortic stiffness, measured by cf-PWV, in patients with RHT and moderate/severe OSA, but treatment may prevent its progression, in contrast to no-CPAP therapy., please discuss and cite doi:10.1111/jsr.12990.
- long-term outcomes in osa treatment adherence depend on different covariabilil. Although it is established in the literature that obstructive sleep apnea and nasal surgery are related by high nasal resistance, which reduces compliance with cpap treatment, clinicians do not take this parameter into consideration, and often the patient does not tolerate cpap, please discuss and cite doi:10.23812/19-522-L-4.
- Patients with daytime sleepiness had a more severe OSA and could presente a greater arterial stiffness improvement after CPAP therapy, independently from age and BP. Besides sleepiness, cf-PWV reduction after CPAP therapy could be associated to CV risk factors, and less to sleep study parameters. please discuss and cite doi: 10.1186/s12890-017-0518-z.
Author Response
We sincerely thank the reviewer for his/her thoughtful comments and constructive suggestions. We have revised our manuscript in light of these recommendations and completed our manuscript with new information, as detailed below.
- Obstructive sleep apnea (OSA) is associated with cardiovascular co-morbidities and mortality. Arterial stiffness is an independent predictor of cardiovascular risk and mortality, and is influenced by the presence of OSA and related comorbidities. There is a paucity of data regarding long-term evolution of arterial stiffness in CPAP-treated OSA patients., please discuss and cite doi:10.1371/journal.pone.0236667.
We briefly mentioned arterial stiffness in OSA patients and completed our reference list accordingly.
Although medium-term (6 months) PAP use can prevent further arterial stiffening in patients with OSA and resistant hypertension [63], a 7.5-year follow-up observational study showed that PAP compliance does not prevent arterial stiffness progression in obese OSA patients [64].
- please better explain study protocol via a flow-chart
We modified figure 1 in order for it to better illustrate the study protocol.
- In the adherence to treatment of the patient with obstructive sleep apnea, mood or behavioral disorders of the patient are molot relevant to consider. Indeed, it has been shown how these disorders are already present in the child and therefore should be well investigated. Likewise, the disorders are reduced with surgical or cpap therapy., please discuss and cite doi:10.3390/children8100921
We briefly mentioned OSA behavioral effects in pediatric patients and completed our reference list accordingly.
Decreased PA can also be attributed to anxiety and depression [73]. Repetitive hypoxia causes neural injury, fatigue, impaired concentration and loss of libido [81]. This explains behavioral changes in pediatric OSA patients [82] and that impaired QoL, depression and anxiety, which frequently coexist and potentiate each other in adult OSA patients [57,73,81], are partially reversed by PAP [81].
-please adopt the adequate guidelines to improve the research protocol according to the equator database.
We revised our manuscript according to the STROBE cheklist and the equador guidelines for observational studies and mentioned this in the ”study design section”
Our protocol is compatible with the Strengthening the Reporting of Observational Studies in Epidemiology (STROBE) checklist [45] and the EQUATOR guidelines for observational studies [46].
We modified the title and added the specific study type in the abstract. We also referenced our previous manuscripts that provide further information regarding our initial cohort, study protocol and preliminary results.
Data regarding our cohort, study protocol and preliminary (2 month) results have been previously published [33–37].
We clearly defined the study objective
Our objective was to evaluate long-term adherence to pharmacological and non-pharmacological treatment in overweight patients with moderate-severe OSA and hypertension. and to evaluate changes in weight, sleepiness and quality of life.
We also underlined our study limitations
The study is limited by its few participants and potentially low statistical power, which limits generalizability of our results. Another important limitation is the collection of follow-up data via telephone, which raises the concern regarding the validity of patient reported clinical data (weight status, blood pressure, heart rate) and questions the correct assessment of PAP adherence (hours/night), which was essential for the study group categorization (adherent versus non-adherent).
- Resistant hypertension (RHT) is associated with obstructive sleep apnea (OSA) and increased aortic stiffness, measured by carotid-femoral pulse wave velocity (cf-PWV). An interesting study evaluated in a randomized controlled trial, the effect of Continuous positive airway pressure (CPAP) treatment on cf-PWV in comparison with a control group in patients with RHT and moderate-severe OSA. One-hundred and sixteen patients were randomized to 6-month CPAP treatment (56 patients) or no therapy (60 patients), while keeping their antihypertensive treatment unchanged. Carotid-femoral pulse wave velocity was performed at the beginning and end of the 6-month period. Intention-to-treat intergroup differences in cf-PWV changes were assessed by a generalized mixed-effects model with the allocation group as a fixed factor and adjusted for age, sex, changes in mean arterial pressure and the baseline cf-PWV values. In conclusion, a 6-month CPAP treatment did not reduce aortic stiffness, measured by cf-PWV, in patients with RHT and moderate/severe OSA, but treatment may prevent its progression, in contrast to no-CPAP therapy., please discuss and cite doi:10.1111/jsr.12990.
We briefly discussed arterial stiffness in OSA patients and completed our reference list accordingly.
Although medium-term (6 months) PAP use can prevent further arterial stiffening in patients with OSA and resistant hypertension [63], a 7.5 year follow-up observational study showed that PAP compliance does not prevent arterial stiffness progression in obese OSA patients [64].
- long-term outcomes in osa treatment adherence depend on different covariabilil. Although it is established in the literature that obstructive sleep apnea and nasal surgery are related by high nasal resistance, which reduces compliance with cpap treatment, clinicians do not take this parameter into consideration, and often the patient does not tolerate cpap, please discuss and cite doi:10.23812/19-522-L-4.
We briefly mentioned nasal resistance as a cause for PAP discontinuation and completed our reference list accordingly.
Elevated nasal resistance increases expiratory discomfort and lowers PAP adherence, but is usually overlooked in clinical practice [52].
- Patients with daytime sleepiness had a more severe OSA and could presente a greater arterial stiffness improvement after CPAP therapy, independently from age and BP. Besides sleepiness, cf-PWV reduction after CPAP therapy could be associated to CV risk factors, and less to sleep study parameters. please discuss and cite doi: 10.1186/s12890-017-0518-z.
We briefly discussed the association between ESS scores, AHI and PAP-associated benefits.
The presence of daytime sleepiness is a predictor of severe apnea and of a higher clinical benefit with PAP use [56].
We hope that you find our responses satisfactory and that we have managed to improve the quality of our manuscript,
Sincerely,
All authors